# Plastic hepatocyte states limit liver cancer development

Lauren S. Strathearn[1,9], Yuki Hayata[1,2,9], Abhinav Illendula [1,3], Charles K. Hewett[1], Mingjia Chen[1], Guoshun He[1,3], María Escribano-Cebrián [1,4], Brianna Jarboe [1,5], Laura Gómez-Tomé[1,4], Nicola de Prisco[1], Michael Slifker[6], Satoshi Kawamura[2], Hayato Nakagawa [2], David Rossell[7], Ben Z. Stanger [8] & Joan Font-Burgada [1] ✉

The liver has remarkable regenerative capacity owing to the boundless proliferative potential of hepatocytes. During liver injury, sustained regeneration must be balanced by mechanisms limiting overgrowth and tumorigenesis. Epithelial plasticity is frequently observed during liver damage and is thought to mediate production of biliary epithelial cells (BECs) or hepatocytes, depending on tissue needs. Here we show that hepatocytes persisting in plastic states are present in virtually all liver injury contexts, representing the predominant outcome of hepatocyte reprogramming rather than their full BEC conversion. By developing tools to trap mouse hepatocytes in plastic states in vivo and using models of regeneration and transplantation, we show that plastic hepatocytes are refractory to proliferation cues from the microenvironment. Unlike terminally differentiated hepatocytes, plastic hepatocytes resist proliferation driven by endogenous oncogenic stimuli. Thus, acquisition of plastic states represents a protective mechanism that constrains hepatocyte proliferation, limiting overgrowth and tumorigenesis during liver disease.

Chronic inflammatory conditions that impair tissue integrity are closely associated with cancer risk. The mechanisms promoting tumorigenesis are well established[1], however, how damaged tissues balance the need for tissue regeneration while actively implementing strategies to suppress tumorigenesis is not understood. Cellular plasticity, whereby cells switch identity through cellular reprogramming mechanisms (e.g., dedifferentiation, transdifferentiation), is a process particularly active during tissue injury and is thought to be important for efficient regeneration. However, dysregulation of this plasticity has also been linked to aging and cancer development[2,3].

In the liver, chronic tissue damage and regeneration are strongly associated with cancer development[4,5]. In these conditions, a well-established paradigm of cellular plasticity exists between the two epithelial cell types, hepatocytes and biliary epithelial cells (BECs)[6,7]. Genetic lineage tracing has shown that BECs can reprogram into hepatocytes in models mirroring human end stage liver disease, where hepatocyte proliferation is severely compromised or absent[8–11]. In contrast, hepatocyte reprogramming to BEC is prominent in cholestatic injuries[12] and can even restore the intrahepatic biliary tree in a mouse model of Alagille Syndrome[13]. However, hepatocyte reprogramming is not limited to models with biliary injury, having been observed using

[1]Cancer Signaling and Microenvironment Program, Fox Chase Cancer Center, Philadelphia, PA, USA. [2]Department of Gastroenterology and Hepatology, Graduate School of Medicine, Mie University, Tsu, Japan. [3]Lewis Katz School of Medicine, Temple University, Philadelphia, PA, USA. [4]Faculty of Experimental Science, Francisco de Vitoria University, Madrid, Spain. [5]Drexel University College of Medicine, Philadelphia, PA, USA. [6]Biostatistics and Bioinformatics Facility, Fox Chase Cancer Center, Philadelphia, PA, USA. [7]Department of Economics, Universitat Pompeu Fabra, Barcelona, Spain. [8]Abramson Family Cancer Research Institute, Division of Gastroenterology, Department of Medicine, University of Pennsylvania Perelman School of Medicine, Philadelphia, PA, USA. [9]These authors contributed equally: Lauren S. Strathearn, Yuki Hayata. ✉e-mail: joan.font-burgada@fccc.edu

lineage tracing in various types of liver damage[12]. Notably, this process is characterized by low efficiency, with only a small fraction of hepatocytes reaching full transdifferentiation, resulting in a spectrum of plastic cells with intermediate features of both hepatocytes and BECs, in a morphologic and gene expression continuum[12,14].

Importantly, plastic hepatocytes displaying intermediate hepatocyte–BEC states are commonly identified in human liver disease specimens, suggesting that the emergence of these cells is a conserved response to liver injury[12,15]. However, much focus has been placed on plasticity's contribution to producing terminally differentiated cells, while the potential significance of cells presenting intermediate plastic states remains largely unexplored.

Here we show that partial hepatocyte reprogramming, defined by hepatocytes persisting in plastic states, is a widespread and abundant response across diverse liver disease models, representing a fundamental outcome of hepatocyte plasticity in response to liver damage, rather than the generation of additional biliary epithelial cells. To address the lack of tools for studying these intermediate states, we develop genetic systems to capture hepatocytes in partial states of reprogramming in vivo. Remarkably, hepatocytes in plastic states resist both extrinsic proliferation cues and intrinsic oncogene-driven growth. This proliferation restraint is conserved across experimental models and contexts, uncovering a universal property of hepatocyte plasticity. Together, these findings support a model in which plastic states are inherently cytostatic in hepatocytes, balancing regenerative demands with tissue size control and thereby acting as a tumour suppressive mechanism.

## Results

### High prevalence of hepatocytes in plastic states across models of liver injury

To estimate the prevalence of hepatocytes in plastic states, we performed a systematic immunofluorescence analysis of a panel of mouse models of liver injury. These included, toxin-induced damage ($CCl_4$ and DDC), genetic ($Fah^{-/-}$ and MUP-uPA) and surgical (PHx and BDL) with DDC and BDL being cholestatic models (Fig. 1a). HNF4α, a hepatocyte lineage marker was independently paired with three markers of hepatocyte reprogramming: Hes1 a canonical readout of Notch activity, since this pathway is essential for this process[12], as well as Sox9 (early) or Osteopontin (intermediate), bona fide markers of hepatocyte to BEC reprogramming[12,16].

As expected, normal control mouse livers had a very small proportion of HNF4α+ hepatocytes displaying Hes1 and Sox9[17]. In contrast, all mouse models of liver injury displayed a significantly higher proportion of hepatocytes with concomitant expression of Hes1, Sox9 or Osteopontin (Opn) (Fig. 1a and Supplementary Figs. 1, 2). Strikingly, the abundance of hepatocytes in plastic states (Hes1+ or Sox9+) increased proportionally with the severity of liver damage, exceeding 50% of all hepatocytes in BDL and $Fah^{-/-}$ models, compared to ~10% in the $CCl_4$ model. Opn+ hepatocytes appeared less abundant compared to Sox9 or Hes1 in all models, with the highest frequencies in cholestasis (BDL and DDC). This observation indicates that hepatocyte reprogramming is a bottleneck with progressively fewer hepatocytes found at advanced stages of transdifferentiation, a process that is favoured in the presence of bile duct damage. Hepatocytes in plastic states were not evenly distributed within the tissue, showing a strong bias toward the portal area, especially for Opn+ hepatocytes (Supplementary Figs. 1, 2).

Taken together, these results indicate that partial hepatocyte reprogramming is a universal response to liver damage and regeneration, predominantly giving rise to hepatocytes in plastic states, referred to hereafter as plastic hepatocytes. Notably, this outcome occurs even in cholestatic injury models, where hepatocyte reprogramming was previously assumed to primarily yield fully differentiated BECs.

### Developing a transposon-based system to capture hepatocytes in plastic states in the normal liver

To further study the properties of plastic hepatocytes in a controlled setting we next developed an in vivo system to stably entrap hepatocytes in plastic states within a normal liver background, enabling their study while minimizing confounding factors from underlying tissue damage and inflammation. Notch signalling is essential for BEC differentiation during development[18] as well as for hepatocyte to BEC transdifferentiation[12]. Most importantly, ectopic activation of Notch signalling in hepatocytes is sufficient to drive their reprogramming to BEC without overt oncogenesis[12,19]. Capitalizing on these mechanistic insights, we designed a PiggyBac transposon vector, CAG > NICD-tdT, to constitutively express the Notch intra-cellular domain (NICD) together with a tdTomato (tdT) cassette for the genetic labelling of hepatocytes targeted by the transposon (Fig. 1b). Delivery of this transposon through hydrodynamic tail vein injection (HDTVi) in C57BL/6 wildtype mice generated sparse single cell tdT+ hepatocytes with robust NICD and Hes1 expression levels (Supplementary Fig. 3a). Expectedly, as early as 1 week after injection, nearly all tdT+ hepatocytes lost expression of the hepatocyte lineage marker HNF4α (>90%) and acquired the expression of Sox9 and Opn (>98%) (Fig. 1c, d). Only after 3 weeks, a small proportion of tdT+ cells (2%) expressed the mature BEC marker Cytokeratin-19 (CK19), which substantially increased (~12%) at 6 weeks after HDTVi (Fig. 1c, d), with some of them forming biliary structures (Supplementary Fig. 3b). Morphologically, these marker expression changes were accompanied by a striking reduction in cell size (Fig. 1e). Approaching normal BEC cell size was associated with expression of the CK19 marker (Supplementary Fig. 3c). The clonal nature of the experimental approach left the liver tissue unaltered (Supplementary Fig. 3d). A CAG > tdT control transposon (Fig. 1b) generated tdT+ labelled hepatocytes with expected normal hepatocyte marker expression (100% HNF4α+ and CK19-, 99% Sox9- and Opn-) and normal hepatocyte cell size (Fig. 1c–e).

Towards trapping hepatocytes in plastic states while avoiding full BEC transdifferentiation, we exchanged the CAG promoter for the hepatocyte-specific Transthyretin (TTR) promoter (Fig. 1b). Since the TTR promoter activity is specific to hepatocyte cell identity[20], we reasoned that in each cell, NICD levels would be linked to their transdifferentiation state. Therefore, NICD activity should be reduced when cells transition closer to BEC identity, while highest when more hepatocytic. Thus, by dynamically restricting NICD expression levels during reprogramming in vivo, we could potentially maximize the number of cells that persistently remain in plastic states between hepatocytes and BECs. Indeed, in mice injected with a TTR > NICD-CAG>tdT transposon, approximately half of tdT+ hepatocytes completely lost HNF4α expression. Concurrently, around 70% of tdT+ cells had high levels of Sox9 and Opn with very few (<3%) showing detectable CK19 levels at 6 weeks (Fig. 1c, d). Furthermore, the average size of tdT+ cells was intermediate compared to CAG > NICD-tdT and CAG > tdT controls (Fig. 1c, e).

Noticeably, between one and three weeks after TTR > NICD-CAG > tdT transposon injection, there was a time-dependent increase in the proportion of HNF4α+/tdT+ cells and concomitant decrease of Opn+/tdT+ cells, accompanied with average cell size gain (Fig. 1d, e). These observations indicated that the TTR promoter initially induced stronger expression of NICD to drive a transcriptional BEC program surge, which gradually lost strength once the TTR promoter was unable to sustain high levels of NICD expression, effectively capturing hepatocytes in plastic states. This gradual loss of reprogramming was accentuated when cells were analysed at 12 weeks after transposon injection with no tdT+ hepatocytes found to be CK19+ (Supplementary Fig. 3e, f).

To further analyse the TTR promoter activity dynamics, we coupled two independent cassettes into the same transposon, one (CAG > NICD-YFP) for driving full reprogramming and the other

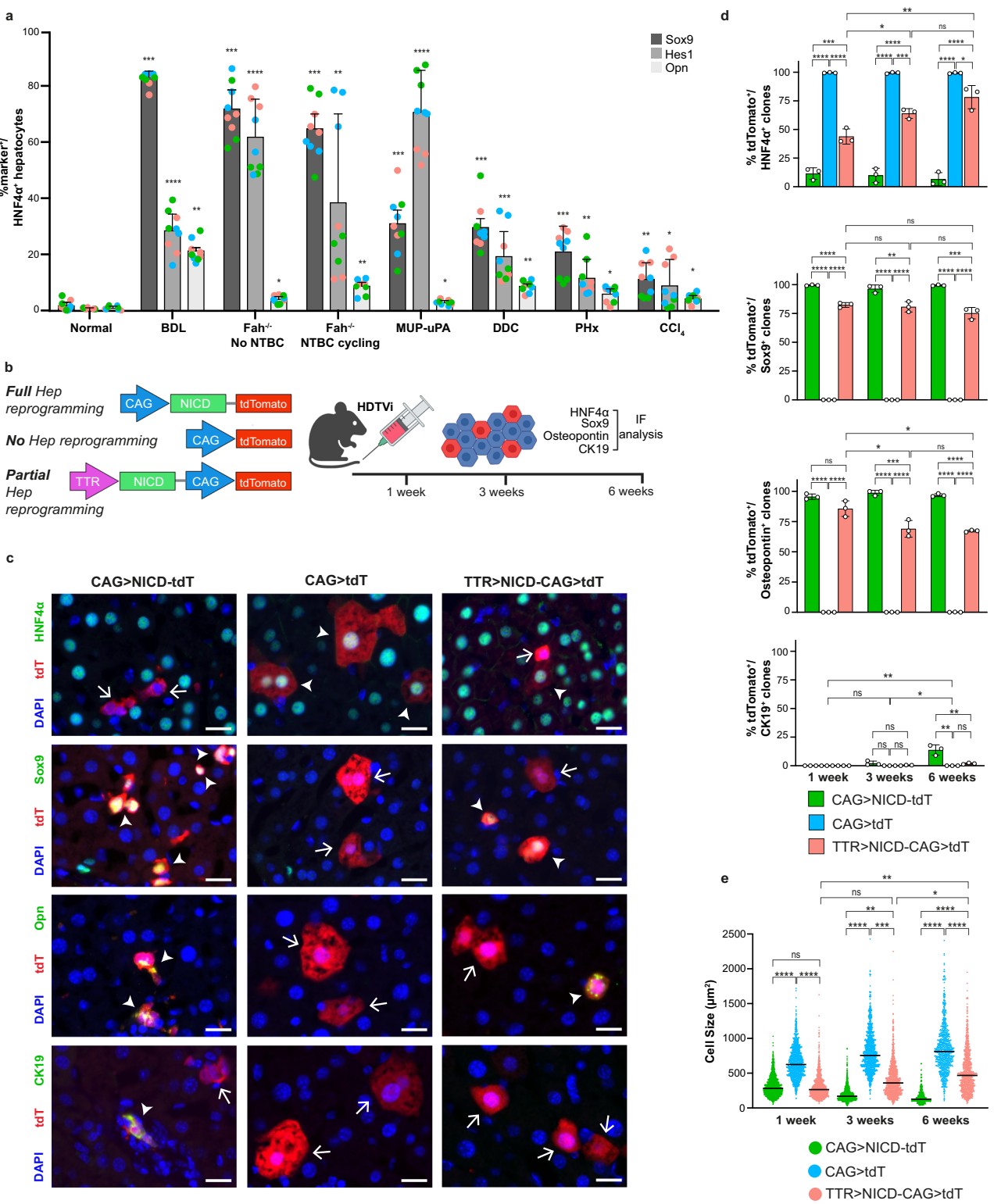

(TTR > tdT) to report TTR promoter activity (Supplementary Fig. 4a). A group of C57BL/6 mice were injected with this transposon and YFP/tdTomato fluorescence was measured at 1,3 and 6 weeks. Over time tdT intensity decreased significantly compared to YFP intensity, as hepatocytes were reprogramming (Supplementary Fig. 4b, c), confirming that TTR promoter activity decreases especially as hepatocytes transition closer to BEC identity.

Based on the observation that endogenous plastic hepatocytes are preferentially located near the portal area (Supplementary Figs. 1, 2),

we re-analysed the data in Fig. 1d, e considering the spatial zonation of the liver. Hepatocytes differ substantially in metabolic profiles across the portal to central vein axis. This divergence is based on different concentrations of nutrients, metabolites and hormones that drive differential expression patterns in hepatocytes. To perform this analysis, we relied on two well-established markers, glutamine synthase (GS) and E-Cadherin (E-Cad) to histologically define three zones. E-Cad+ hepatocytes delimiting periportal zone 1, GS+ hepatocytes marking pericentral hepatocytes in zone 3 and double negative hepatocytes

**Fig. 1 | Prevalence of plastic hepatocytes during liver damage and the development of a transposon-based induction tool. a** Expression of hepatocyte-BEC reprogramming markers Hes1, Sox9 and Osteopontin (Opn) in HNF4α⁺ hepatocytes was quantified in the livers of mice subjected to the indicated damage models. $n = 3$ mice per injury. Data displayed are means ± SD. Coloured dots designate three random areas quantified from the same mouse sample. Each damage model was compared to normal by an unpaired two-tailed $t$-test with Benjamini–Hochberg adjustment. ****$p < 0.0001$; ***$p < 0.001$; **$p < 0.01$; *$p < 0.05$. BDL bile duct ligation. CCl₄ chronic carbon tetrachloride, DDC 0.1% 3,5-diethoxycarbonyl-1,4-dihydrocollidine diet. Fah⁻/⁻ NTBC cycling, Fumarylacetoacetate hydrolase⁻/⁻ mice cycled off Nitisinone three times. Fah⁻/⁻ no NTBC, Nitisinone withdrawn from Fah⁻/⁻ mice continuously for 3 weeks. MUP-uPA, transgenic mice expressing urokinase-type plasminogen activator under control of the Major Urinary Protein promoter. PHx, 70% partial hepatectomy. **b** Schematic illustrating the experimental strategy to induce hepatocytes at various stages of reprogramming within the normal liver

using transposons. TTR transthyretin, NICD Notch-intracellular domain. Created in BioRender. Hewett, C. (2026) https://BioRender.com/xjlpzpi. **c** Representative images of immunofluorescence for tdTomato (tdT) in combination with either HNF4α (hepatocyte marker), Sox9 and Opn (early BEC markers), and CK19 (mature BEC marker) on liver sections 3-weeks post-HDTVi of the indicated transposons. Arrowheads denote marker positive tdT⁺ clones; arrows denote marker negative tdT⁺ clones. n = 3. Scale bars, 10μm. DAPI, 4′,6-diamidino-2-phenylindole. Quantification of the percentage of tdT⁺ hepatocytes expressing the indicated cell identity markers (**d**) and tdT⁺ cell size (**e**) at 1-, 3- or 6-weeks post-HDTVi of the indicated transposons. In (**d**) data displayed are means ± SD $n = 3$ mice. In (**e**) each data point represents a single cell, and the mean of the data for each transposon/time point from three mice is denoted by solid black lines; $n = 9541$ cells. Statistical significance was determined by one-way ANOVA with post-Hoc Tukey's test for multiple comparisons. ****$p < 0.0001$; ***$p < 0.001$; **$p < 0.01$; *$p < 0.05$; ns not significant.

defining zone 2 in between (Supplementary Fig. 5a). When considering zonation, zone 1 and 2 tdT⁺ hepatocytes induced by the CAG > NICD-tdT transposon presented with similar hepatocyte and BEC marker frequency and size across time points, however, zone 3 tdT⁺ cells had lower rates of HNF4α loss and no expression of CK19, which was associated with a slight increase in average cell size (Supplementary Fig. 5b, c). TTR > NICD-CAG > tdT transposon zonated data revealed a clear phenotypic gradient with the highest frequency of BEC markers in zone 1, decreasing in zone 2 and lowest in zone 3. HNF4α showed an inverse pattern, and cell size changes correlated with levels of marker expression as expected (Supplementary Fig. 5b, c). This differential reprogramming across zones was not caused by biased TTR promoter activity (Supplementary Fig. 6a) or differential transposon integration (Supplementary Fig. 6b, c). In addition, a similar transposon with an independent hepatocyte specific promoter (TBG)[21] produced similar zonated effects (Supplementary Fig. 7). These results suggest that on average, hepatocytes closer to the portal tract (Zone 1) are more responsive to Notch-driven reprogramming compared to hepatocytes surrounding the central vein (Zone 3).

### Entrapped plastic hepatocytes in the normal liver are indistinguishable from endogenous plastic hepatocytes in the damaged liver

To further characterize the plastic hepatocytes generated with the TTR > NICD-CAG > tdT transposon, single cell RNA sequencing was performed on a mixture of tdT⁺ cells, tdT⁻ hepatocytes and tdT⁻/EpCAM⁺ native BECs isolated 3 weeks post-HDTVi from 2 mice (Fig. 2a). Displaying Krt19 (BEC marker) and Scd1 (hepatocyte marker) gene expression levels on a UMAP plot of the data revealed distinct BEC and tdT⁻ hepatocyte populations connected by a continuum of tdT⁺ expressing cells (Fig. 2b). Cells expressing tdT were vastly enriched in the intermediate cluster (94.7% of all tdT⁺ cells) while few tdT⁺ cells (<1.2%) clustered with BECs and only 5.1% still clustered with normal hepatocytes (Supplementary Fig. 8a). Previously reported bulk RNA-seq and scRNAseq data of endogenous hepatocytes reprogramming to BECs in the context of biliary damage demonstrated a gradient of gene expression changes across the hepatocyte to BEC continuum[14]. Primary BEC transcription factors, namely Sox9 and Sox4, are expressed early on, with a decreasing number of cells expressing secondary markers such as Igfbp7 and Cd24a and an even smaller subpopulation, the closest to native BECs, expressing tertiary markers including Krt19, Epcam or Ezr[14]. Analysis of the same collection of markers within our dataset showed that tdT⁺ cells displayed a remarkably similar expression pattern as described for endogenous plastic hepatocytes in the context of liver damage (Supplementary Fig. 8b). In addition, hepatocyte markers such as Alb, Hnf4α or Ttr showed a progressive decrease in expression (Supplementary Fig. 8c).

To determine the overall degree of similarity between entrapped plastic states and those found during endogenous hepatocyte

reprogramming in the context of liver damage[14], we integrated the corresponding scRNAseq datasets. Scd1 and Krt19 expression patterns similarly identified hepatocyte and BEC clusters with a remaining cluster of cells connecting both populations in the integrated data set as was described in the original study (Fig. 2c). Strikingly, analysis of tdT expression revealed that tdT⁺ hepatocytes almost exclusively overlapped with endogenous hepatocytes in plastic states (Fig. 2d), indicating that at the whole transcriptome level both populations were highly similar. Accordingly, correlation analysis revealed a high degree of similarity in gene expression between both plastic hepatocyte populations (Fig. 2e). In addition, we performed GSEA on the entrapped plastic hepatocytes scRNAseq data. In this case, we employed bulk RNAseq data of normal versus endogenous plastic hepatocytes during biliary damage[14] to compute the top differentially expressed genes at various cut-offs to define the gene sets for the analysis. Results showed that all gene sets tested were remarkably enriched for both up- and downregulated genes, with only a few genes not following the expected trends (Fig. 2f and Supplementary Figs. 9, 10).

Taken together, immunofluorescence and scRNA-seq analyses demonstrate that the TTR > NICD-CAG > tdT transposon system faithfully reproduces physiologic levels of Notch activity, enabling the stable entrapment of hepatocytes across a continuum of plastic states that naturally arise during liver injury. These experimentally induced cells are phenotypically and transcriptionally indistinguishable from their endogenous counterparts, which constitute the predominant outcome of hepatocyte reprogramming in liver disease, validating the system as a physiologically relevant model for studying hepatocyte plasticity in vivo.

### Plastic hepatocyte states are associated with diminished proliferative capacity during liver regeneration

Establishing the means to entrap hepatocytes in plastic states in the normal liver paved the way to determine their contribution to liver regeneration. Since hepatocyte self-duplication is the major force driving liver regeneration[20,22–24], we designed an experimental setting to simultaneously measure clonal proliferation of two independent populations of hepatocytes. We employed the 70% partial hepatectomy (PHx) model since it induces minimal cellular damage while mobilizing all cells in the liver for proliferation to recover the lost mass. To clonally compare normal with plastic hepatocytes, ROSA26^EYFP/+ mice, which carry a Cre-dependent reporter, were infected with AAV-TBG > iCre, where the thyroxine-binding globulin (TBG) promoter ensures exclusive Cre expression in hepatocytes. Low doses of AAV-TBG > iCre were used to sparsely label normal YFP⁺ hepatocytes. One week after infection, the same mice were HDTV injected with the TTR > NICD-CAG>tdT transposon to induce plastic states in tdT⁺ hepatocytes. Three weeks post-HDTVi, 70% PHx was performed, and the excised left and median lobes were collected as pre-PHx samples serving as baseline for clonal size. Mice were allowed to recover for an

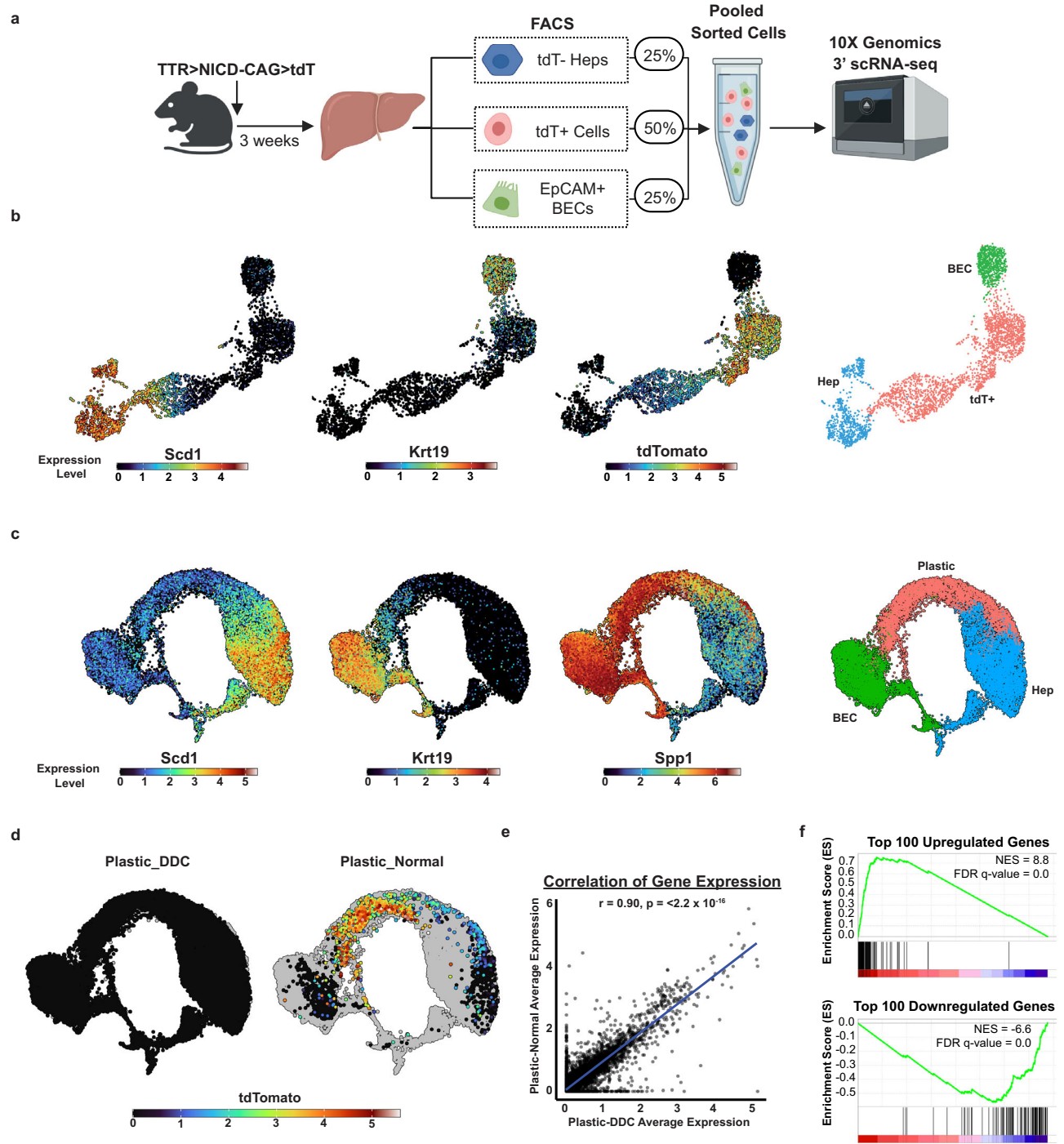

**Fig. 2 | Expression profiles of entrapped plastic hepatocytes are indistinguishable from endogenous plastic hepatocytes. a** Schematic illustrating experimental design to generate single cell transcriptomes of tdT⁺ intermediate cells, tdT⁻ normal hepatocytes and tdT⁻/EpCAM⁺ native BECs from normal livers using gating strategies outlined in Supplementary Figs. 18, 19. This experiment was performed with two independent C57BL/6 mice. Created in BioRender. Hewett, C. (2026) https://BioRender.com/lto5015. **b** UMAP plots showing the expression of hepatocyte (Scd1) and BEC (Krt19) markers, as well as tdT following single-cell RNA sequencing as described in (**a**), with gene expression marker-based clustering for hepatocytes (Hep), BEC and tdT⁺ cells. *n* = 3486 cells. **c** UMAP plots showing expression of Scd1, Krt19 (CK19), and Spp1 (Opn), with marker-based clustering of Hep, BEC and plastic hepatocytes (Plastic) of the integrated data from panel **b** (Plastic-Norm) with single-cell RNA gene expression data from the DDC-diet induced damage (Merrel et al.[14]) denoted as Plastic-DDC. *n* = 21420 cells. **d** Left plot shows tdT expression in cells from Plastic-DDC dataset (no expression) while Plastic-Norm cells are in grey. Right panel shows tdT expression in Plastic-Norm while Plastic-DDC cells are displayed in grey. **e** Pearson correlation analysis between the partially reprogrammed hepatocyte populations from Plastic-Normal vs Plastic-DDC datasets. **f** GSEA of the top 100 genes up-or down-regulated in Plastic-DDC bulk RNAseq dataset, analysed on the tdT⁺ vs hepatocyte ranked gene list (x-axis).

additional week to promote liver regeneration (post-PHx sample) (Fig. 3a). We next performed clonal analysis comparing both YFP⁺ and tdT⁺ clones in Pre- vs Post-PHx samples (Fig. 3b). As anticipated, the percentage of normal YFP⁺ hepatocyte clones greater than 1 cell significantly increased in post-PHx samples, with an average of 1.4-fold increase over the pre-PHx samples and similar to what has been previously described[25] (Fig. 3c, d). However, a similar percentage of tdT⁺ clones containing more than 1 cell were found in pre- and post-PHx,

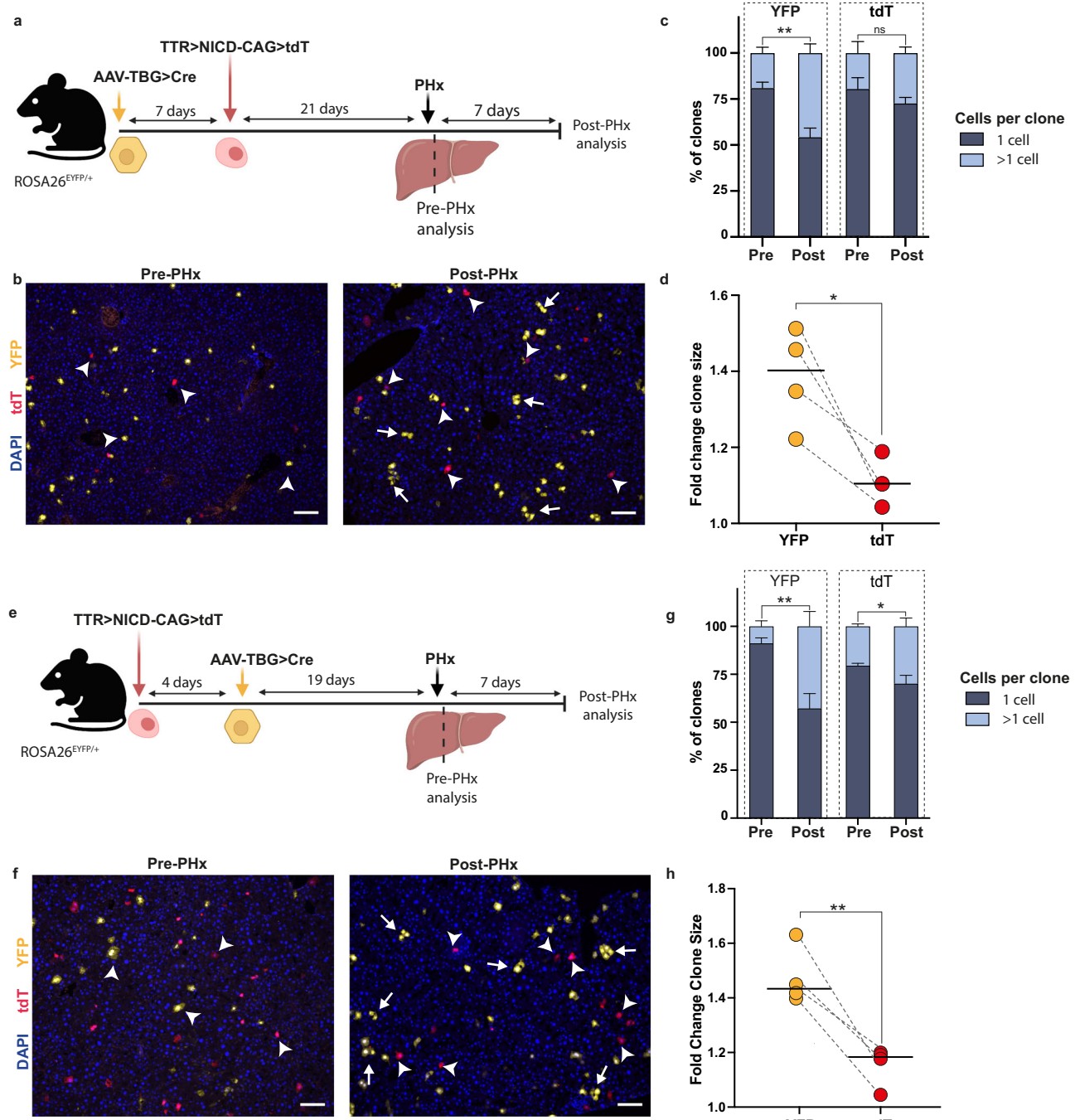

**Fig. 3 | Plastic hepatocyte states limit proliferative response to partial hepatectomy regenerative stimuli. a** Schematic showing experimental design to clonally label and trace normal (YFP⁺) and transposon generated plastic tdT⁺ hepatocytes before (Pre-) and after (Post-) 70% partial hepatectomy (PHx). Created in BioRender. Hewett, C. (2026) https://BioRender.com/kvh56eq. **b** Representative images of immunofluorescence analysis for YFP⁺ (normal heps) and tdT⁺ (plastic hepatocytes) pre-and post-PHx. Arrows denote proliferating clones; arrowheads denote single cell clones. Scale bars, 100 μm. **c** Quantification of the number of cells per YFP⁺ and tdT⁺ clone was performed on pre- and post-PHx samples. Data shown is mean ± SD; $n = 4$ mice. Statistical significance was determined by paired, two-tailed $t$-test. **d** the fold change in clone size from the same clones quantified in (**c**) is shown. Lines denote the mean of the data. Datapoints collected from the same mice are connected by dashed lines. $n = 4$ mice. Paired, two-tailed $t$-tests were performed to determine statistical significance. **e** Schematic of alternative experimental design in which the order of HDTV and AAV injections were switched from (**a**). Created in BioRender. Hewett, C. (2026) https://BioRender.com/cobu9ji. **f**–**h** Analysis of experiment and statistics (**e**) as described in (**b**–**d**). Scale bars, 100 μm. $n = 4$ mice. **$p < 0.01$; *$p < 0.05$; ns not significant.

with only 1.1-fold increase from pre-PHx, indicating that hepatocytes in plastic states proliferated markedly less when receiving analogous proliferating signals compared to normal hepatocytes (Fig. 3b–d). Quantification of reprogramming markers in post-PHx samples showed that the reprogramming state of the tdT⁺ population was largely unaltered during the experimental period (Supplementary

Fig. 11a). Because the two clonal populations were generated sequentially, potentially generating confounding effects, we performed the same experiment but reversed the order in which both populations were generated. Results confirmed that hepatocytes displaying plastic states have a reduced capacity to proliferate compared to normal hepatocytes when receiving the same proliferative signalling input

within the same tissue context (Fig. 3e–h). Additionally, when the data was analysed by zone, no major differences in clonal growth were observed (Supplementary Fig. 11b–e).

We next performed transplantation experiments using the Fah[−/−] mouse model of hereditary tyrosinemia[26]. These mice lack a functional fumarylacetoacetate hydrolase (*Fah*) gene, but provision of NTBC in drinking water is sufficient to protect them from disease. Withdrawal of NTBC results in extensive liver damage, liver failure and mouse death. However, transplantation of Fah[+] donor hepatocytes can rescue these mice by efficiently repopulating the injured livers[27,28]. This model provides a very strong selective proliferative pressure towards transplanted Fah[+] hepatocytes and therefore is ideal to measure the intrinsic proliferation capacity of a given hepatocyte population. Plastic hepatocytes (tdT[+]) and normal hepatocytes (tdT[−]) were isolated from donor C57BL/6 mice 3 weeks after injection of the TTR > NICD-CAG>tdT transposon and transplanted into Fah[−/−] recipient mice. After 3 cycles of NTBC withdrawal, mice were sacrificed for clone size analysis (Fig. 4a). Fah[+]/tdT[+] clones presented with a very much reduced size in comparison to Fah[+]/tdT[−] clones derived from normal hepatocytes (Fig. 4b). Quantitative analysis showed that on average, clones derived from plastic hepatocytes were only 12% of the size of those derived from normal hepatocytes (Fig. 4c and Supplementary Fig. 12a). Notably, most Fah[+]/tdT[+] clones retained their plastic phenotype (Fig. 4d and Supplementary Fig. 12b).

To rule out the possibility that plastic hepatocytes could engraft less efficiently, thereby driving the diminished observed growth, we implemented an inducible strategy for generating plastic states through partial reprogramming. Using the Tet-off system LAP>tTA[29] combined with TRE > NICD[30], we generated LAP > tTA;NICD mice. In this setting, tTA is expressed under the hepatocyte specific *Cebpb* gene promoter, tightly suppressing NICD expression when doxycycline (Dox) is added in the drinking water. Upon Dox removal, NICD expression is activated in hepatocytes inducing their reprogramming, though, as with the transposon system, expression is expected to decrease as hepatocyte identity and LAP promoter activity fades.

To compare clonal growth of normal hepatocytes with that of hepatocytes in plastic states induced after transplantation, we injected a transposon expressing tdT (CAG > tdT) in LAP > tTA;NICD donor mice and a transposon expressing YFP (CAG > YFP) into LAP-tTA control donor mice. Both groups were maintained on Dox to suppress NICD expression in the LAP > tTA;NICD donors. After 10 days, hepatocytes were isolated from both groups, sorted to obtain tdT[+] and YFP[+] hepatocytes, mixed and transplanted at 50% into two immunodeficient Fah[−/−] (FRG) mice (Fig. 4e). One mouse remained on Dox to prevent reprogramming in tdT[+] hepatocytes while the other remained on Dox for only 4 days post-transplantation to ensure comparable engraftment, after which Dox was removed to allow partial reprogramming in the tdT[+] population (Fig. 4e). Furthermore, to modulate reprogramming during repopulation, Dox was added mid-regeneration once (Dox off +1), or twice (Dox off +2) (Fig. 4e) in two additional groups of mice.

As observed with the transposon system, plastic states induced in hepatocytes post-transplantation resulted in diminished proliferation capacity (Fig. 4f–i and Supplementary Fig. 13a,c,e). Notably, mice with additional Dox during repopulation produced less reprogrammed tdT[+] clones (Supplementary Fig. 13b, d, f). While these clones still showed reduced proliferation, there was a trend towards more proliferation, significantly closer to normal hepatocytes, as reprogramming was further suppressed (Dox off +2) (Fig. 4i).

Together, these results establish that hepatocytes in plastic states remain refractory to proliferation even in the context of competitive advantage, indicating that plastic states are inherently cytostatic and do not contribute to generate cellular mass during liver damage. Consistently, analysis of proliferation markers in the single-cell RNA-seq dataset from Fig. 2, shows that proliferation is restricted to terminally differentiated BECs or hepatocytes during damage and regeneration with no detectable proliferation activity in plastic hepatocytes (Supplementary Fig. 14).

## Hepatocytes in plastic states are resistant to oncogene-induced proliferation and tumorigenesis

Chronic liver disease (CLD) induced cirrhosis represents the greatest risk factor for the development of liver cancer[4,5]. Therefore, reduced proliferation capacity linked to hepatocytes in plastic states could have profound implications for liver carcinogenesis given the high prevalence of these plastic hepatocytes in the diseased liver (Fig. 1a). Hepatocellular carcinoma (HCC) is the most common type of liver cancer followed by intrahepatic cholangiocarcinoma (iCCA)[31]. While HCC is assumed to originate exclusively from hepatocytes, mouse models have demonstrated that iCCA can arise not only from BECs[32] but also from mouse and human hepatocytes[19,33,34], indicating that hepatocyte reprogramming could contribute to tumorigenesis. To determine whether hepatocytes in states of plasticity are intrinsically resistant to proliferation and therefore less likely to contribute to tumorigenesis, we employed three oncogenes— myristoylated-*Akt1* (AKT*), *Kras*[G12D], and *Ccnd1*—each known to induce hepatocyte proliferation and HCC development within 7–10 months[35–38]. Using transposons, we delivered these oncogenes with consistent dosage into C57BL/6 wildtype mice, coupled with the induction of full reprogramming (CAG > NICD-PGK>Oncogene-tdT), partial reprogramming (TTR > NICD-PGK>Oncogene-tdT) or no reprogramming (PGK>Oncogene-tdT) (Fig. 5a).

As previously reported[19,33], NICD overexpression with a strong ubiquitous promoter combined with AKT*, induced visible tumours with histological features consistent with CAA, such as CK19 positivity and cuboidal cells forming glandular structures (Fig. 5b and Supplementary Fig. 15a). Also, consistent with previous studies[35], AKT* expression alone in normal hepatocytes resulted in lipid accumulation, increased cell size, and clonal proliferation (Fig. 5c) without producing visible tumours at 12 weeks. In contrast, hepatocytes in plastic states (TTR > NICD-PGK > AKT*-tdT) showed minimal clonal growth with no signs of tumorigenesis (Fig. 5d). Quantitative analysis confirmed that plastic hepatocytes formed significantly smaller clones compared to both normal hepatocytes and fully reprogrammed hepatocytes (Fig. 5e). This finding aligns with our observations in regeneration models, where hepatocytes expressing physiological NICD levels, and thus in plastic states, displayed markedly reduced proliferation compared to normal hepatocytes lacking Notch activation. Together, these results demonstrate that diminished proliferative capacity is an intrinsic property of hepatocytes in plastic states, compared to both normal hepatocytes and fully reprogrammed BECs, manifesting during both regenerative stimuli and oncogenic activation.

To rule out oncogene-specific effects, we repeated the same approach with *Ccnd1* and *Kras*[G12D], obtaining similar results: fully reprogrammed hepatocytes formed iCCA-like tumours, normal hepatocytes exhibited clonal expansion, and plastic hepatocytes had reduced growth compared to the other two cellular states (Fig. 5f–m and Supplementary Fig. 15b, c).

Histological analysis of TTR > NICD-PGK>Oncogene-tdT samples further showed a zonation-dependent proliferative phenotype (Supplementary Fig. 16a, c, e). In zones 1 and 2, tdT[+] hepatocytes showed minimal clonal growth, with the exception of a few clones with CK19[+] cells displaying BEC morphology, which exhibited moderate clonal expansion. In contrast, in zone 3, tdT[+] hepatocytes predominantly retained a hepatocytic phenotype, with some clones showing limited growth near central veins. Quantitative analysis confirmed that partial reprogramming was associated with reduced clonal growth across the three liver zones although oncogenes induced more proliferation when clones were located in Zone 3 (Supplementary Fig. 16b, d, f). These results align with previous observations that TTR > NICD-

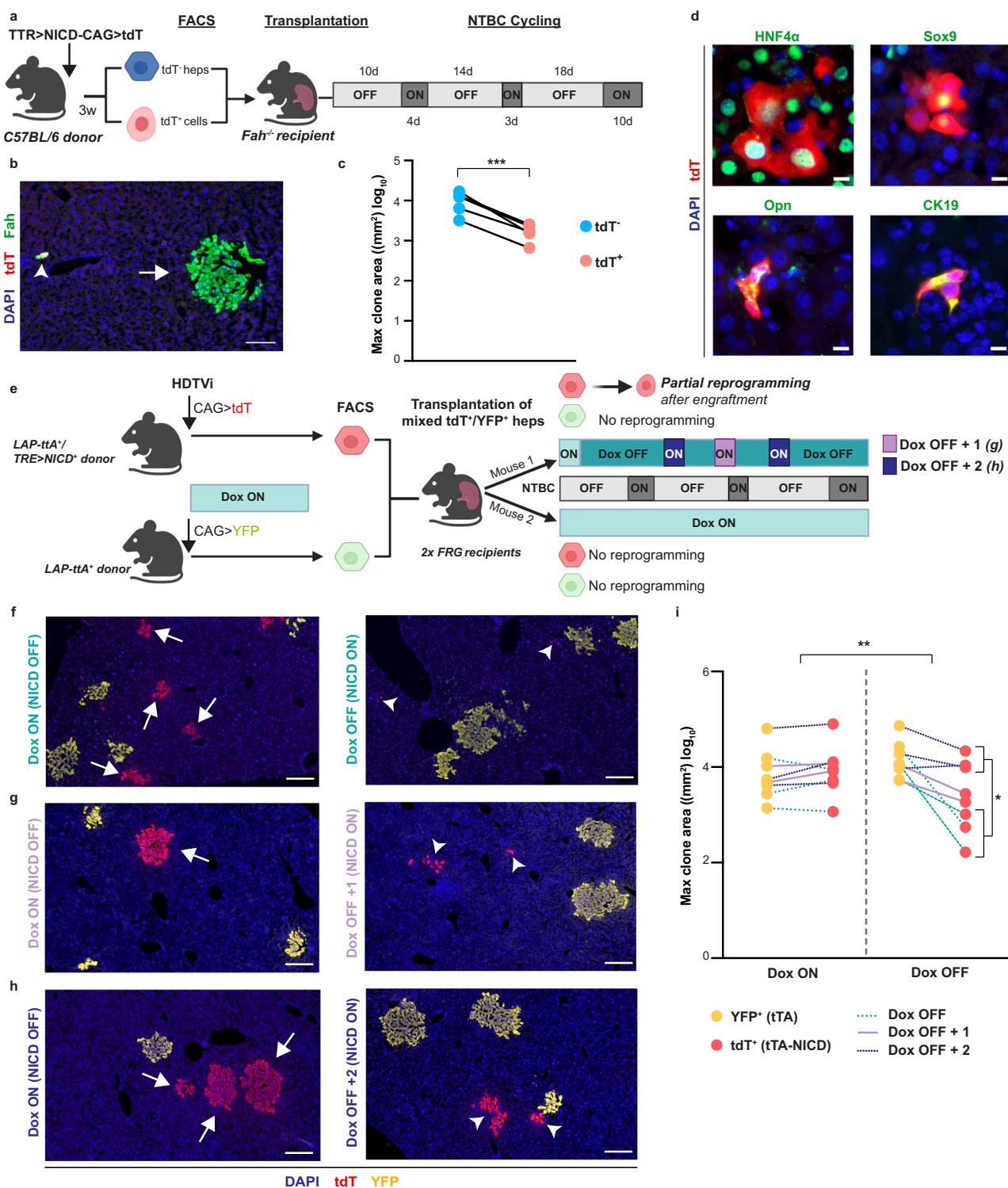

CAG>tdT transposon primarily induces partial reprogramming, while only few hepatocytes escape reprogramming or fully reprogram (~5% and ~1% respectively, Supplementary Fig. 8a), and that zone 3 hepatocytes are less susceptible to Notch-induced reprogramming (Supplementary Fig. 5). Collectively, these findings suggest that hepatocytes escaping plastic states during reprogramming in the TTR > NICD-PGK>Oncogene-tdT model selectively proliferate with a zonation bias, more biliary in zone 1 and more hepatocytic in zone 3.

Finally, to directly test whether the lack of oncogene-driven proliferation in plastic hepatocytes depends on their identity state, we employed Alb-Cre; Sox9$^{F/F}$ mice, in which hepatocyte-BEC

reprogramming is strongly suppressed[39] (Supplementary Fig. 17a, b). We injected the TTR > NICD-PGK>Oncogene-tdT transposons in both Alb-Cre; Sox9$^{F/F}$ and control Sox9$^{F/F}$ mice. Consistent with results in C57BL/6 mice, no visible tumours were detected in control Sox9$^{F/F}$ mice for any of the oncogenes at 12 weeks (Supplementary Fig. 17c, e, f). Remarkably, in Sox9-deleted livers, clonal growth of hepatocytic cells was apparent for all three oncogenes, with especially extensive proliferation in the AKT* group, where visible tumours were observed (Supplementary Fig. 17). Analysis at earlier time points showed that the TTR > NICD-PGK > AKT*-tdT transposon efficiently promoted hepatocytic clonal proliferation across liver zones when

**Fig. 4 | Plastic hepatocytes have diminished proliferation capacity under competitive advantage. a** Schematic illustrating experimental set-up to compare repopulation potential of tdT⁻ (normal) hepatocytes and tdT⁺ (plastic) hepatocytes and sorted using the gating strategies outlined in Supplementary Figs. 18, 20. Created in BioRender. Hewett, C. (2026) https://BioRender.com/drjmjid. **b** Representative immunostaining of liver sections from Fah⁻/⁻ transplanted mice stained with Fah and tdT antibodies. Arrow: Fah⁺/tdT⁻ clone (normal); arrowhead: Fah⁺/tdT⁺ clone (plastic). Scale bar, 100 μm. **c** Plot showing mean maximum area of tdT⁻ or tdT⁺ clones across 10 serial sections. Connected dots: values from the same mouse. n = 5. Statistical significance was determined by paired, two-tailed t-test. ***p < 0.001. **d** Immunostaining of liver sections from the same mice showing HNF4α, Sox9, Osteopontin and CK19 in tdT⁺ clones. Scale bars, 10 μm. **e** Schematic illustrating experimental set-up to compare repopulation potential of YFP⁺ (normal) hepatocytes and tdT⁺ plastic hepatocytes (partial reprogramming induced after engraftment). Reprogramming in the tdT⁺ population was prevented by treating donors and transplanted mice with Doxycycline (Dox). Four days after

transplantation one mouse of the pair was withdrawn from Dox to induce reprogramming. To further manipulate reprogramming levels, Dox was re-administered during the repopulation phase: removed throughout (**f**), added back for 3 days halfway through (**g**) or added back twice for 3 days every 20 days (**h**). FRG, Fah⁻/⁻; Rag2⁻/⁻; Il2rg⁻/⁻. Created in BioRender. Hewett, C. (2026) https://BioRender.com/drjmjid. **f**–**h** Representative immunofluorescence images for tdT and YFP of livers from FRG recipients transplanted as in (**e**). Arrows: normal hepatocytic tdT⁺ clones; arrowheads: reprogrammed tdT⁺ clones. Scale bars, 100 μm. **i** Mean maximum area of tdT⁺ and YFP⁺ clones across 10 serial sections from eight pairs (Dox ON, Dox OFF) of FRG recipients mice transplanted, n = 8 (**e**). Each pair of connected dots are the two values (YFP⁺, tdT⁺) from the same mouse. For each mouse, we computed the log-difference between YFP and tdT + . First, an unpaired t-test was used to compare the n = 8 log-differences in Dox ON vs. the n = 8 in Dox OFF. **p < 0.01. Second, within the Dox OFF group, differences between Dox pulsing groups were assessed by one-way ANOVA with post-Hoc Tukey's test for multiple comparisons. *p < 0.05.

---

hepatocytes were unable to transition into plastic states (Supplementary Fig. 17d). Together, these findings establish that the intrinsic resistance to proliferation observed in plastic hepatocytes is not simply due to Notch activation but fundamentally linked to the maintenance of a plastic state.

## Discussion

Epithelial plasticity during liver damage and disease has long been recognized as an important phenomenon in both human and animal models[15,40]. The dominant view has been that epithelial reprogramming serves to replenish the epithelial compartments in need: biliary epithelial cells (BECs) reprogram into hepatocytes when hepatocyte proliferation is impaired, and hepatocytes reprogram into new BECs during cholestasis. However, evidence suggests that this model may be an oversimplification. For instance, hepatocyte reprogramming has been shown to be inefficient; even in cholestatic injuries, only a small fraction of hepatocytes fully transdifferentiate into terminally differentiated BECs[14]. Additionally, hepatocyte reprogramming has been found in models without bile duct injury, indicating that this process occurs more broadly than previously thought[12].

In this study, we show that the main outcome of Notch-induced hepatocyte reprogramming is partial reprogramming, producing hepatocytes that persist in plastic states. Across the spectrum of models we analysed, at least 10% of all hepatocytes showed evidence of partial reprogramming with some reaching as high as 80%. Moreover, this is most likely an underestimation, since our analysis scored double positivity with the hepatocyte marker HNF4α, therefore hepatocytes in more advanced stages of reprogramming which lose HNF4α expression were not accounted for. In addition, we found that plastic hepatocytes arising from partial reprogramming were present across all tested models of liver injury, although with varying frequencies. This widespread occurrence is consistent with previous isolated observations, such as the high Hes1/HNF4α frequency seen in metabolic dysfunction-associated fatty liver disease (MAFLD) and metabolic dysfunction-associated steatohepatitis (MASH)[41] and the co-expression of Sox9/HNF4α in several mouse models[12,42] as well as lineage-tracing data showing that full conversion of hepatocytes into BECs is exceedingly rare[12,14].

Studying such an abundant and diverse population of cells, with dynamic identities and transcriptomes, poses significant challenges. Conventional lineage tracing tools fall short in their ability to capture plastic states in vivo, as they can only label a subset of states at a time, and further changes in cellular identity could confound analysis. To overcome these limitations, we developed a system that simultaneously induces and traps hepatocytes in plastic states in vivo, minimizing reversion to hepatocytes or full reprogramming into BECs. By leveraging the TTR promoter, our model limits NICD expression to physiologic levels that mirror those observed during endogenous liver injury, unlike

prior models that used strong ubiquitous promoters to drive supraphysiologic NICD levels, resulting in full reprogramming into BECs which has been shown to be highly infrequent during liver disease[12,14].

Single-cell RNA-seq analysis of these entrapped plastic hepatocytes in the normal liver confirmed their striking similarity to endogenous plastic hepatocytes found in diseased livers, consistent with prior findings that transcriptomic differences among BECs, reprogramming hepatocytes, and normal hepatocytes are minimally influenced by damage-associated changes[14]. Additionally, we observed a gradient of reprogramming susceptibility, with periportal (zone 1) hepatocytes being most susceptible and pericentral (zone 3) hepatocytes the least, suggesting that distinct epigenetic landscapes associated with zonation influence hepatocyte amenability to Notch-driven reprogramming.

By using this system, we uncovered that hepatocytes in plastic states are inherently refractory to proliferation, even when they have a competitive advantage. This fundamental feature of a hepatocyte population so abundant in diseased livers raises an important question: what evolutionary advantage might non-proliferative plastic hepatocytes confer when the liver is in urgent need of regeneration? Clues to this apparent paradox emerged when plastic hepatocytes were challenged with oncogenic stimuli. In contrast to normal hepatocytes and fully reprogrammed BEC-like cells, which readily underwent clonal expansion, plastic hepatocytes consistently resisted oncogene-driven proliferation.

The role of Notch in cancer is well known to be highly context-dependent, acting either as an oncogene or a tumour suppressor depending on cell type, tissue, or signalling context[43,44]. Although Notch activity has previously been described as oncogenic in hepatocytes[45], our results collectively demonstrate that Notch is not intrinsically oncogenic. Rather, its effect is determined by dose-dependent changes in cellular identity. At physiological levels, as seen during liver injury, Notch induces cytostatic plastic states in hepatocytes that suppress proliferation. In contrast, supraphysiological Notch activation drives full reprogramming into BEC-like cells, which become competent to proliferate and, in the presence of oncogenes, give rise to cholangiocarcinoma. This highlights a narrow threshold between tumour-suppressive and tumour-promoting outcomes of Notch signalling in hepatocytes.

Taken together, these findings suggest that one path toward HCC development may involve circumventing the cytostatic plastic states in hepatocytes during liver damage, either through suppression of Notch-induced reprogramming or by cooperation with specific oncogenes or tumour suppressors that override it. Indeed, a substantial proportion of human HCC tumours show evidence of Notch pathway activation[45]. Conversely, other oncogenic combinations may push plastic hepatocytes past the high threshold for full reprogramming, promoting biliary conversion and ultimately contributing to CCA formation. These findings offer mechanistic support for our broader

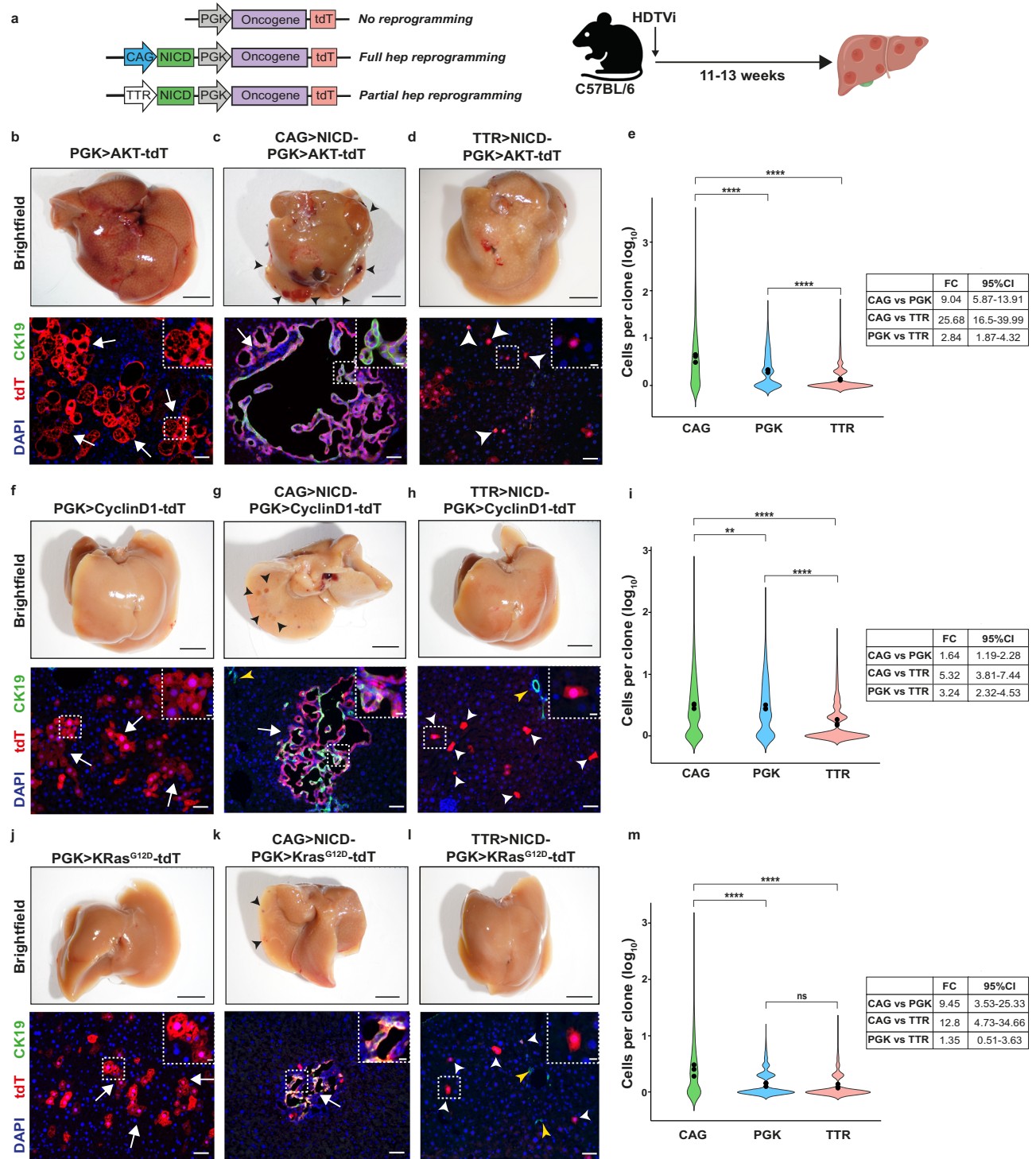

**Fig. 5 | Plastic hepatocyte states limit oncogene-induced proliferation and tumorigenesis. a** Diagram illustrating the experimental design to compare oncogene-induced proliferation in different populations of hepatocytes depending on their reprogramming state. Created in BioRender. Hewett, C. (2026) https://BioRender.com/uc2ipoy. **b–d** C57BL/6 mice 12 weeks post-HDTVi of the indicated myristoylated-AKT (AKT*) expressing transposons. Shown are whole liver photos and representative images of immunofluorescence of liver sections from the same mice for tdT and CK19 ($n = 3$). Black arrowheads indicate visible tumours. White arrows show examples of proliferating tdT+ clones. White arrowheads show examples of tdT+ single cell clones. Scale bars whole livers, 0.5 cm; Scale bars immunofluorescence: main, 50 μm; inset, 10 μm. **e** Violin plot showing the tdT+ clone size (cells per clone) generated by the indicated transposons, 8 weeks post-HDTVi. Dots represent the mean value from individual mice ($n = 3$). Statistical analysis was performed by fitting a negative binomial model that featured random

effects for each mouse and an additive fixed effect for the three groups (CAG/PGK/TTR). Statistical significance of overall differences between the three groups with a likelihood-ratio test revealed a significant effect of transposon type on clone size ($p < 0.0001$). Subsequently, pairwise comparisons between groups were conducted using estimated marginal means with Tukey adjustment for multiple comparisons. Fold change (FC) and 95% confidence intervals (CI) are shown for the designated comparisons. $n = 3945$ clones. **f–h** C57BL/6 mice 13 weeks post-HDTVi of the indicated *Ccnd1*-expressing transposons. Data provided are as described in (**b–d**). Yellow arrowheads indicate normal endogenous BECs. **i** Analysis of clone size from the same mice in (**f–h**) as in (**e**). $n = 4830$ clones. **j–l** C57BL/6 mice 11 weeks post-HDTVi of the indicated *Kras*G12D-expressing transposons. Data provided are as described in (**b–d**). **m** Analysis of clone size and statistics from the same mice in (**j–l**) as described in (**e**). $n = 4211$ clones. ****$p < 0.0001$; **$p < 0.01$; ns not significant.

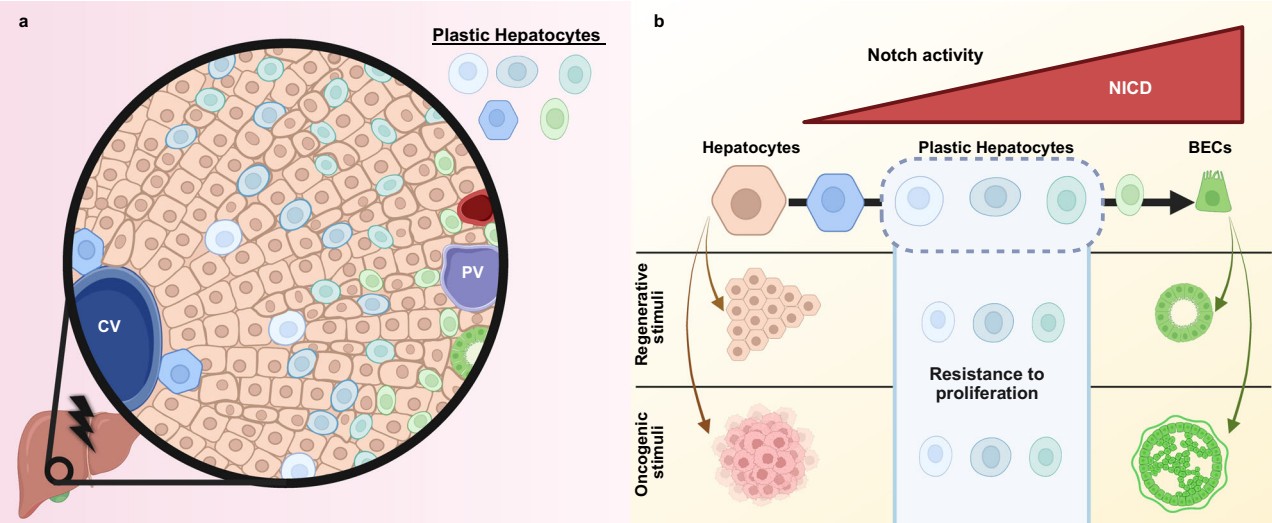

**Fig. 6 | Plastic hepatocyte states limit liver cancer development.** Graphical depiction of the findings of this study. Briefly, in (**a**) liver injury induces a pervasive and zonally-biased emergence of plastic hepatocytes through partial reprogramming to BECs. **b** Utilising a transposon to express physiological levels of NICD, hepatocytes at various stages of partial reprogramming can be generated which are indistinguishable from endogenous plastic hepatocytes. Robust cytostasis was identified in plastic hepatocytes during liver regeneration, translating to an enhanced tumour suppressive response even in the presence of sustained oncogene activation. In contrast, hepatocytes and BECs at each end of the reprogramming spectrum are licensed to proliferate during regeneration, and initiate tumorigenesis following oncogene activation. Hep, hepatocyte. BEC, biliary epithelial cell. NICD, Notch intracellular domain. Created in BioRender. Font-Burgada, J. (2026) https://BioRender.com/ostt981.

model: that Notch-induced plastic states represent a cytostatic checkpoint during liver injury, temporarily halting proliferation in a way that balances regeneration with overgrowth and tumorigenesis. Interestingly, the high prevalence of hepatocytes in cytostatic plastic states observed in severely damaged livers (such as BDL or Fah$^{-/-}$, Fig. 1a) may represent a maladaptive consequence of this otherwise protective mechanism, potentially contributing to the markedly low hepatocyte proliferation rates characteristic of advanced liver disease.

In summary (Fig. 6), our findings show that partial hepatocyte reprogramming, resulting in the accumulation of plastic hepatocytes, is a prevalent and significant adaptive response during liver injury, acting as an independent pathway from the production of new BECs. By developing a system to study hepatocytes in plastic states, we have uncovered an inherent cytostatic effect of physiologic Notch-induced plasticity. We propose that zonally-biased partial hepatocyte reprogramming is a fundamental mechanism that limits unrestrained proliferation helping to balance tissue regeneration, overgrowth and cancer suppression in the context of liver damage where growth factors and inflammatory cytokines are in high levels. While our work is based on mouse models, further studies will be required to establish whether similar plastic states that have been observed in human liver disease exert comparable cytostatic effects and whether this process can be manipulated to enhance regeneration without promoting carcinogenesis. Notably, Notch signalling has been implicated in driving metaplastic transitions in other tissues[46–48], most prominently in acinar-to-ductal metaplasia in the pancreas[49]. Whether Notch-induced plastic states in other tissues similarly confer tumour-suppressive properties, as observed in the liver, remains an important question for future research.

## Methods

### Mice
Animal experiments were performed according to protocols approved by Fox Chase Cancer Center Institutional Animal Care and Use Committee (IACUC). Unless specified, mice were maintained on standard laboratory mouse chow and drinking water ad libitum in a 12-hour light-cycle, temperature, and humidity-controlled room. All mice were obtained from their referenced sources. Fah$^{-/-}$ [50] and *Fah*$^{-/-}$; *Rag2*$^{-/-}$; *Il2rg*$^{-/-}$ (FRG)[28] mice were maintained on Picolab High Energy Mouse Diet (Labdiet; #5LJ5) and 8 mg/L CuRx™ Nitisinone (NTBC) (Yecuris; #20-0027) ad libitum. LAP>tTA[29]; TRE > NICD[48] mice were provided with 200 mg/ml doxycycline-supplemented drinking water ad libitum to prevent activation of NICD. MUP-uPA mice[51] were sacrificed at 5-weeks old for analysis of hepatocyte reprogramming. C57BL/6 mice (Jackson Laboratory; #000664) were treated with 0.1% 3,5-Diethoxycarbonyl-1,4-Dihydrocollidine (DDC) diet for 3 weeks. For carbon tetrachloride (CCl$_4$)-induced liver damage C57BL/6 mice were treated with 0.5 μl/gr CCl$_4$ by intraperitoneal injection, every 3 days for a total of 12 doses. Mice between 8-9 weeks were used for experiments. Male mice were used for experiments using C57BL/6, ROSA26$^{tdTomato/+}$, ROSA26$^{EYFP/+}$ and Fah$^{-/-}$. Alb-Cre (Jackson Laboratory, #003574) and Sox9$^{F/F}$ [51] mice were bred together. Male and female mice were used for experiments in the Alb-Cre; Sox9$^{F/F}$; LAP-tTA-TRE-NICD; FRG background. No gender bias was observed. Mice that harboured tumours were monitored daily and sacrificed if tumour burden comprised >10% of the mouse's body weight, the mouse lost 20% of its body weight after injection, or if the mouse exhibits signs of distress or pain. No mice in this study exceeded these limits.

### Plasmid and AAV vectors
All piggyBac transposon and vectors were constructed by Vectorbuilder. Notch intracellular domain (NICD) sequence was obtained from Addgene Plasmid #26891[52]. For cloning of myristoylated AKT expressing vectors, pT3-myr-AKT-HA was obtained from Xin Chen via Addgene (Plasmid #31789)[53]. The hyperactive piggyBac transposase vector was obtained from Vectorbuilder. For mice experiments, transposons (30μg) and transposase (15 μg) were diluted together in saline and injected intravenously by HDTVi[54]. The total injection volume was equivalent to 10% of mouse body weight i.e. a 25 g mouse received 2.5 ml of DNA solution. For partial hepatectomy experiments in ROSA26$^{EYFP/+}$ mice, AAV(DJ8)-TBG>Cre was obtained from Vector Biolabs (VB1724) and injected through the tail vein in PBS at 1 ×10$^{10}$ genome copies per mouse.

## Tissue collection, processing and imaging

Mouse livers were collected following intracardiac Zinc-formalin perfusion. Livers were incubated overnight in Zinc-Formalin for further fixation, followed by PBS washing and incubation with 100 mM Tris (pH9.4), 10 mM DTT for 2 h. Livers were then incubated overnight in 15% sucrose, then 30% sucrose before embedding in Tissue-Tek®OCT (Sakura, #4583). For immunofluorescence staining, 8 µm liver tissue sections were cut from frozen blocks and collected on Fisherbrand™ Superfrost™ Plus Microscope Slides. Slides were incubated in citrate buffer at 96 °C for 1 h for antigen retrieval, followed by PBS washing and 20 min incubation in 0.1%Triton-X100-PBS. After further PBS washing, tissue sections were incubated in blocking buffer (2% Donkey serum-0.1% Tween-PBS) for 30 min. Primary Antibodies were diluted in blocking buffer and incubated with tissue sections overnight at 4 °C. Antibodies used in immunofluorescence were selected based on validations from their respective vendors detailed in Supplementary Table 1. Slides were washed 3X in 0.1%Tween-PBS and incubated with the appropriate secondary antibodies at room temperature (RT). After 2 h, slides were washed 3X with 0.1%Tween-PBS and 2X with PBS. Slides were then quickly washed in water and 70% ethanol before incubating for 20 min in 0.1% Sudan Black. Slides were washed extensively in 0.02%Tween-PBS to remove Sudan Black excess, and then tissue sections were incubated with DAPI (1 µg/ml) for 10 min at RT for nuclear staining. Tissue sections were then mounted using Mowiol prior to imaging. Slides were viewed and images captured on a Zeiss AxioObserver, and images were processed in Zeiss Zen Software. A Nanozoomer S60 instrument (Hamamatsu) was used for whole slide scanning. Tyramide signal amplification (TSA) staining (Thermo Fisher Scientific; B40933) was performed in three experiments. Firstly, to amplify the YFP signal in partial hepatectomy experiments. Secondly, to analyse the zonation of tdT⁺ clones a panel of four antibodies was required and TSA staining enabled the use of antibodies of the same species. Third, for the analysis of liver damage models it was necessary to enhance the signal of the Sox9, Osteopontin, and Hes1 signals. TSA staining was performed first according to manufacturer's instructions, followed by a second antigen retrieval step before continuing with the routine immunofluorescence staining protocol described above. For damage model staining, several changes were made to the staining protocol. For Hes1 staining only, sections were cut at 5 µm. The first antigen retrieval in citrate buffer was done using a pressure cooker for 45 minutes at 7.5 psi. Primary antibody staining was carried for 1 hour at room temperature instead of overnight. For Sox9 staining only, no Sudan Black was used.

## Primary mouse liver cell isolation

To isolate hepatocytes, tdT⁺ cells and BECs for single cell RNA sequencing, mouse livers were perfused via the portal vein at 37 °C. Briefly, livers were first perfused with 50 ml of Ca²⁺/Mg²⁺-free HBSS supplemented with 10 mM Hepes and 0.5 mM EGTA at a rate of 8 ml/minute. This was followed by perfusion with Ca²⁺/Mg²⁺-containing HBSS supplemented with 10 mM Hepes and 4 mg/ml Liberase TM (Sigma, #5401127001) at 8 ml/minute. The second step of perfusion was continued until the liver showed clear signs of digestion inside the liver capsule. The liver was extracted from the mouse and the gallbladder removed. The livers were placed in 15 ml Ca²⁺/Mg²⁺-free HBSS and the liver capsule was gently broken to release digested liver cells. The cell suspension was filtered through a 70 µm cell strainer and centrifuged at $50 \times g$ for 1 min with no deceleration. The resulting pellet comprised hepatocytes and the supernatant the remaining liver cell populations. The hepatocyte pellet was washed in Ca²⁺/Mg²⁺-free HBSS a second time followed by a third wash in Ca²⁺/Mg²⁺-free HBSS supplemented with 20 µg/ml DNAse (Alfa Aesar, J62229MB). The hepatocyte pellet was then resuspended in 40% Percoll-Williams media and centrifuged at $150 \times g$ for 5 min to pellet viable hepatocytes. Hepatocytes were resuspended in William's media and stained with

1 µg/ml DAPI prior to Fluorescence Activated Cell Sorting (FACS). The supernatant from the first centrifugation step enriched for non-hepatocyte populations was centrifuged at $1000 \times g$ for 5 min to pellet cells and then resuspended in Ca²⁺/Mg²⁺-free HBSS supplemented with 20µg/ml DNAse. The cell suspension was centrifuged at $1000 \times g$ for 5 min and the pellet resuspended at a concentration of 10 million cells per ml in 2%FBS-DMEM. Cells were incubated with 10 µg/ml Mouse BD Fc Block (BD, #553141) for 5 min on ice, followed by a 15-min incubation with 2.5 µg/ml anti-EpCAM-FITC on ice in the dark. Cells were washed twice in PBS and resuspended in 2% FBS-2mM EDTA-PBS with 1 µg/ml DAPI prior to FACS.

To isolate hepatocytes for intrasplenic transplantation into FRG recipients, mouse livers were perfused via the inferior vena cava (IVC) at 37 °C. Two-step perfusion was performed as above however flow rate was reduced to 4 ml/min. Hepatocytes were pelleted and washed as described above. For the final centrifugation step, hepatocytes were resuspended in 30% Percoll to pellet viable cells. Hepatocytes were prepared for FACS as above.

Cell sorting for single cell RNA-sequencing was done using a BD Aria II with a 100 µm nozzle. Hepatocyte sorting for intrasplenic transplantation was carried out on a BD Symphony S6 instrument fitted with a 100 µm nozzle.

## Mouse surgical procedures

For all surgical procedures, mice were anesthetized with 5% isoflurane and placed on a heating pad for the length of the surgery. Skin around the surgical site was disinfected with betadine and 70% ethanol. Bupivacaine (4 mg/kg) was provided intraperitoneally as pre-surgery analgesia. Two-thirds partial hepatectomy was performed as described[55]. For Fah repopulation experiments ~60,000 hepatocytes resuspended in 15% FBS/DMEM were intra-splenically transplanted as previously described[17]. Immediately post-surgery the NTBC in the drinking water was completely removed for a total of 10 days. Mice were then returned to NTBC drinking water for a 4-day recovery period. A second cycle of consisted of 14 days removal and 3 days recovery. NTBC was removed again 18 days, followed by a 10-day recovery period before sacrifice. For FRG repopulation experiments, ~100,000 hepatocytes were transplanted and recipients were subjected to the same NTBC cycling regimen as above. In this experiment, FRG recipients were pre-treated with 50 µg/ml dox-supplemented drinking water for 2 days prior to transplantation, and both recipients remained on dox until 4 days-post transplantation. Within each pair of FRG recipients transplanted, one remained on dox throughout, while the other was removed from dox for either the full length of the experiment, or the majority of the experiment. Dox re-addition lasted for 3 days. Bile duct ligation was performed as described in Tag et al.[56] and mice were sacrificed 2.5 weeks post-surgery.

## Single cell RNA-sequencing

Preparation of cells for single cell RNA-sequencing (scRNA-seq) was performed using the Chromium Next GEM Single Cell 3′ GEM, Library & Gel Bead kit (10x Genomics, PN-10000121). Post-FACS, cells were scanned on a Zeiss Axio observer to obtain cell count and confirm >85% viability and presence of single cells. Sorted cell populations were pooled together and loaded into a single well of a Next GEM Chip G at the following ratios: 25% tdT⁻ hepatocytes, 25% EpCAM⁺/tdT⁻ BECs and 50% tdT⁺ cells. Manufacturer's instructions were followed to partition cells into Gel Beads-in-emulsions (GEMs) together with unique 10x barcodes using the Chromium Controller. cDNA libraries were produced from GEMs to generate a barcoded single cell library for each mouse. Sequencing data were obtained using the Illumina NovaSeq 6000 PE150 platform.

Raw sequencing data files were first processed using the Cell Ranger pipeline from 10× Genomics using default parameters. Reads were mapped to the GRCm39 reference modified to include the

tdTomato sequence, and the resultant data were then processed using the Seurat V5.1.0 package in R V4.2.1. Cells which did not meet QC criteria (n_feature count between 400–7000 and <15% mitochondrial genes) were filtered out prior to downstream analysis. The subsequent data were subjected to uniform manifold approximation and projection (UMAP) dimensionality reduction for visualization. Data visualization was performed using standard Seurat parameters. Markers of hepatocyte, BEC and reprogramming cell identity were used to identify clusters. Immune and endothelial cell clusters were removed, as well as clusters consisting only of cells with low total gene and transcript counts. Figures were generated using SCpubr package[57] in R.

This in house generated scRNA-seq data from transposon-induced plastic hepatocytes was then integrated with scRNA-seq data generated in Merrell et al. Specifically, .h5 files were downloaded from GSE157698. Data was merged in Seurat, and QC analysis selected for cells with n_feature count between 800–4000 and a mitochondrial gene count below <25% across both datasets. The 'rpca' method was used to integrate the data within Seurat. Clustering was performed as described above.

Heatmap and GSEA analysis were performed using bulk RNA sequencing data from Merrell et al. (GSE156894). We used the GSEA-Preranked method of GSEA[58] to assess the concordance between a set of genes identified as differentially expressed between reprogramming and normal hepatocytes in a bulk RNA-seq experiment (GSE156894), and a ranked list of differentially expressed genes between tdT[+] versus normal hepatocyte scRNA-seq data. Default parameters were used for GSEAPreranked, except that enrichment scores were calculated based on the classic method, rather than the default weighted option.

Cell-cycle analysis was carried out with Cyclone (scran) on log-normalized RNA counts using the default mouse marker pairs and parameters[59,60].

UMAP plots describing single-cell spatiotemporal expression profiles for *Ttr*, *Gls2* and *Cyp2e1* were accessed at db.cngb.org/stomics/lista.

The R-studio packages and their versions used for the analysis in this study are as follows: Seurat 5.1.0, SCpubr 2.0.2, scran 1.26.2, stringr 1.5.1, ggplot2 3.5.2, patchwork 1.3.0, dplyr 1.1.4, cowplot 1.1.3, org.Mm.eg.db 3.15.0, SingleCellExperiment 1.20.1, scales 1.3.0.

### Image analysis

Images were analysed manually in ImageJ or NDP.view2. Analyser was blinded to the variable of interest when determining the areas to analyse in every sample but were not blinded to the sample ID. For quantification of reprogramming markers in Fig. 1a, three areas per liver were randomly selected and HNF4α[+] cells were identified. Within this HNF4α[+] population, cells were scored either positive or negative for Sox9, Opn or Hes1 expression. For quantification of cell identity marker expression (Fig. 1/Supplementary Figs. 3, 5, 7, 11, 12, 13, 17) one area per liver lobe was selected at random. Areas varied from 1 mm² to the whole lobe depending on the density of tdT[+] clones in the sample. For analysis of Sox9 and HNF4α expression, only tdT[+] clones with visible nuclear DAPI signal were counted to avoid false negative results. To analyse the zonation of tdT[+] clones, E-Cadherin and GS were used to mark the periportal and pericentral zones, respectively. Clones that spanned multiple zones were excluded in the analysis. To assess TTR promoter activity (Supplementary Fig. 4) liver sections were cut, stained, and scanned with identical exposure settings in parallel. 200 marker positive cells were traced for each mouse in ImageJ. The YFP and tdTomato mean pixel intensity values were then extracted for each cell. To analyse proliferation pre- and post-partial hepatectomy (Fig. 3), ~200 tdT[+] and 200 YFP[+] clones were analysed per lobe within a randomly selected area. For analysis of clone area in Fig. 4, 10 serial sections were analysed across a total depth of 200 μm. Twenty 10 μm sections were cut and every second section was analysed, i.e. sections

1, 3, 5, 7, 9, 11, 13, 15, 17, 19. Every clone was identified throughout the entire tissue section and the area of every clone was measured at each depth. Data is presented as the maximal clone area observed through the 200 μm depth. To quantitate cell identity marker expression in Supplementary Figs. 12, 13, all tdT[+] clones were analysed across 5 serial sections (total depth of 100 μm). For analysis of clone size in Fig. 5 one area per liver lobe was selected at random. Areas varied from 1 mm² to the whole lobe depending on the density of tdT[+] clones in the sample.

### Statistical analysis and reproducibility

Source data are provided as a Source Data file. Data was processed in GraphPad Prism. The specific statistical test performed on each dataset is detailed in the corresponding figure legends. Generally, when comparing 2 groups, a *t*-test was used; for comparisons involving >2 groups, a one-way ANOVA was carried out followed by a post-hoc Tukey's test or Benjamini–Hochberg (as appropriate) for multiple comparisons. Asymmetric data was log(10) transformed to improve normality and stabilize variance.

Statistical analyses of clone size for oncogenic-induced proliferation were performed using R and utilizing the glmmTMB package for model fitting and the emmeans package for post-hoc comparisons. A negative binomial generalized linear mixed model (GLMM) was fitted to the clone size data to account for overdispersion observed in the clone size counts. The model included the transposons (CAG > NICD-PGK > Oncogene, TTR > NICD-PGK-Oncogene and PGK > Oncogene) as a fixed effect as well as each mouse replicate per condition as random effects. The negative binomial model was fit to zero-shifted counts, i.e. subtracting 1 to all counts so that the smallest value is zero, which improved the negative binomial model fit. Both the zero-shifting and presence over-dispersion relative to the Poisson model were supported by the Bayesian information criterion (BIC). Pairwise comparisons were conducted using estimated marginal means with Tukey adjustment for multiple comparisons. For the analysis of the interaction between YFP and tdTomato fluorescence across time points during hepatocyte reprogramming, the approach was similar. In this case, a linear mixed effects model was fitted to the log transform data using the nlme package. The emmeans package was used for post-hoc analysis.

*n* numbers for all experiments are described in figure legends. All experiments were replicated at least 3 times, except for scRNA-seq data and Fah transplantation studies which are representative of 2 individual replicates per experiment. Sample sizes were determined based on our previous experience with the models and published studies in the field. No statistics were used to predetermine the sample size.

### Reporting summary

Further information on research design is available in the Nature Portfolio Reporting Summary linked to this article.

## Data availability

The raw sequencing data generated in this study have been deposited in the GEO database under accession code GSE284235. All source data generated in this study are provided in the Supplementary Information/Source Data file. Source data are provided with this paper.

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

## Acknowledgements

We thank Drs. S. Balachandran, G. Rall, A. Bellacosa, D. Wiest and G. Napolitano for comments. We also thank Dr. Markus Grompe for providing Fah$^{-/-}$ knockout mice and members of his lab, Leslie Wakefield and Dove Enicks, for technical assistance. Technical support was provided by the Flow Cytometry Core, Histopathology and Laboratory Animal Facilities at Fox Chase Cancer Center (NIH P30CA006927). L.S.S. and Y.K were supported by Internal Fox Chase Cancer Center awards via a William J. Avery Postdoctoral Fellowship and a Board of Associates Fellowship, respectively. A.I. was supported by a T32 training grant (T32GM142606). B.Z.S. was supported by the Fred and Suzanne Biesecker Pediatric Liver Center. Research was supported by NIH grants DP2CA258224, R01CA289703 and a W.W. Smith Charitable Fund Award to J.F.-B. Experimental graphic summaries were created with BioRender.com and is licensed under CC BY 4.0.

## Author contributions

L.S.S. and J.F-B. conceived and designed the project; L.S.S., Y.H. and A.I. performed experiments with the assistance of C.K.H., M.C., G.H., M.E.-C., L.G.-T. and N.dP.; B.J. and M.S. performed computational analysis. D.R. provided statistical advice. S.K., H.N. and B.Z.S. provided conceptual input and tools. L.S.S. and J.F-B. wrote the manuscript with all authors contributing to the writing and providing advice.

## Competing interests

The authors declare no competing interests.
