## [Transparent Peer Review file · Nature Communications]

Plastic hepatocyte states limit liver cancer development

Corresponding Author: Dr Joan Font-Burgada

Version 0:

Reviewer comments:

Reviewer #1

(Remarks to the Author)

My concerns had already been all addressed in the previous version of the manuscript. I have no further comments. My only suggestion, would be that maybe now some of the relevant data that is in supplementary figures could be brought to main, as there is no concerns anymore on length. But this is just a suggestion and could be discussed with the Editor, if appropriate.

Regarding Reviewer #2's remaining concerns, the authors have provided convincing data to show the existence of partially reprogrammed cells. Reviewer #2 has a valid point about the relevance to human cancer since the study is performed in mouse models. The authors are advised to further discuss this caveat.

My concerns had already been all addressed in the previous version of the manuscript. I have no further comments. My only suggestion, would be that maybe now some of the relevant data that is in supplementary figures could be brought to main, as there is no concerns anymore on lenght. But this is just a suggestion and could be discussed with the Editor, if appropriate.

Regarding Reviewer #2's remaining concerns, the authors have provided convincing data to show the existence of partially reprogrammed cells. Reviewer #2 has a valid point about the relevance to human cancer since the study is performed in mouse models. The authors are advised to further discuss this caveat.

We thank the Referee for their final comments. Following the Referee's suggestion, we have modified the discussion to include the following:

“Although our work is based on mouse models, further studies will be required to establish whether similar plastic states that have been observed in human liver disease exert comparable cytostatic effects and whether this process can be manipulated to enhance regeneration without promoting carcinogenesis.”